# Transforming Prostate Cancer Care: Innovations in Diagnosis, Treatment, and Future Directions

**DOI:** 10.3390/ijms26115386

**Published:** 2025-06-04

**Authors:** Sanaz Vakili, Iman Beheshti, Amir Barzegar Behrooz, Marek J. Łos, Rui Vitorino, Saeid Ghavami

**Affiliations:** 1Department of Human Anatomy and Cell Science, University of Manitoba College of Medicine, Winnipeg, MB R3E 3P5, Canada; s.vakili11@gmail.com (S.V.); iman.beheshti@umanitoba.ca (I.B.); amir.barzegarbehrooz@umanitoba.ca (A.B.B.); 2Biotechnology Center, Silesian University of Technology, 44-100 Gliwice, Poland; 3Department of Medical Sciences, Institute of Biomedicine iBiMED, University of Aveiro, 3810-193 Aveiro, Portugal; 4Department of Surgery and Physiology, Faculty of Medicine, University of Porto, 4200-319 Porto, Portugal; 5Biology of Breathing Theme, Children Hospital Research Institute of Manitoba, University of Manitoba, Winnipeg, MB R3T 2N2, Canada; 6Faculty of Medicine, Akademia Śląska, Rolna 43, 40-555 Katowice, Poland; 7Paul Albrechtsen Research Institute, CancerCare Manitoba, University of Manitoba, Winnipeg, MB R3E 0V9, Canada

**Keywords:** prostate cancer, diagnosis, artificial intelligence, biomarkers, precision medicine, advanced therapies

## Abstract

Prostate cancer remains a major global health challenge, ranking as the second most common malignancy in men worldwide. Advances in diagnostic and therapeutic strategies have transformed its management, enhancing patient outcomes and quality of life. This review highlights recent breakthroughs in imaging, including multiparametric MRI and PSMA-PET, which have improved cancer detection and staging. Biomarker-based diagnostics, such as PHI and 4K Score, offer precise risk stratification, reducing unnecessary biopsies. Innovations in treatment, including robotic-assisted surgery, novel hormone therapies, immunotherapy, and PARP inhibitors, are redefining care for localized and advanced prostate cancer. Artificial intelligence (AI) and machine learning (ML) are emerging as powerful tools to optimize diagnostics, risk prediction, and treatment personalization. Additionally, advances in radiation therapy, such as IMRT and SBRT, provide targeted and effective options for high-risk patients. While these innovations have significantly improved survival and minimized overtreatment, challenges remain in optimizing therapy sequencing and addressing disparities in care. The integration of AI, theranostics, and gene-editing technologies holds immense promise for the future of prostate cancer management.

## 1. Introduction

Prostate cancer (PCa) remains one of the most prevalent malignancies affecting men worldwide, with significant implications for public health and quality of life [1]. The landscape of PCa care has evolved dramatically over the past decade, driven by innovations in diagnostic techniques, treatment modalities, and our understanding of the disease’s molecular underpinnings [1]. Recent epidemiological data indicate that while incidence rates have stabilized in many developed countries, mortality rates show significant geographic and racial disparities [2]. The introduction of prostate-specific antigen (PSA) screening in the 1980s led to a surge in diagnoses but also raised concerns about overdiagnosis and overtreatment of clinically insignificant cancers [3]. This has spurred efforts to develop more precise diagnostic tools and risk stratification methods to guide treatment decisions [3]. Given these ongoing challenges and the rapid evolution of the field, there is a clear need for a comprehensive and up-to-date review. The unifying theme of this review is “transforming care through innovation”, as recent years have seen major advances in both diagnosis and treatment. Novel diagnostic approaches, including biomarker-based tests and advanced imaging techniques, offer the potential for more accurate detection and characterization of prostate cancer, aiming to reduce unnecessary biopsies and better distinguish between indolent and aggressive disease [4].

Concurrently, treatment paradigms are evolving rapidly. The emergence of targeted therapies, immunotherapies, and precision medicine approaches based on genomic profiling has expanded the therapeutic arsenal for prostate cancer [5]. These advancements are particularly significant for patients with advanced or metastatic disease, where traditional treatments have shown limited efficacy [5].

This comprehensive review aims to synthesize and critically evaluate the latest advancements in prostate cancer diagnosis and treatment. We will explore cutting-edge developments in imaging technologies, such as multiparametric MRI and PSMA-PET, which have revolutionized the detection and staging of prostate cancer. The review will also delve into emerging biomarkers and genomic tests that promise to enhance risk stratification and guide personalized treatment decisions.

In the realm of treatment, we will examine the evolution of surgical techniques, including robot-assisted prostatectomy, and assess their impact on patient outcomes. The review will investigate recent innovations in radiation therapy, such as stereotactic body radiation therapy (SBRT), which offers improved precision and reduced side effects. Additionally, we will explore the expanding landscape of systemic treatments, including novel hormone therapies, immunotherapies, and targeted agents that are reshaping the management of advanced prostate cancer. Furthermore, this paper will address the growing interest in active surveillance strategies for low-risk disease and discuss the potential of alternative treatments as an alternative to traditional therapies. We will also examine ongoing clinical trials and emerging treatment paradigms that promise to improve patient outcomes and quality of life.

## 2. Overview of Prostate Cancer Epidemiology and Current Challenges

As highlighted before, prostate cancer remains a significant global health concern, ranking as the second most common cancer in men worldwide. In 2020, there were 1,414,259 new cases of prostate cancer diagnosed globally, accounting for 7.3% of all cancer cases in men [2,6]. The incidence rates vary significantly across regions, with the highest age-standardized rates observed in Oceania, North America, and Europe, while Africa and Asia have comparatively lower rates [7]. The epidemiology of prostate cancer is characterized by several key factors, such as age, racial disparities, geographic variation, and mortality. In brief, the incidence of PCa increases dramatically with age, with approximately 60% of cases diagnosed in men over 65 years old [7]. African American men have higher incidence and mortality rates compared to other racial groups, with incidence rates of 158.3 per 100,000 men, nearly double that of White men [7]. There is a 190-fold difference in incidence rates between the highest (France, Guadeloupe) and lowest (Bhutan) reporting populations [7]. In terms of mortality, prostate cancer caused 375,304 deaths worldwide in 2020 [6]. Prostate cancer management has six key challenges that demand immediate attention and innovative solutions. Overdiagnosis and overtreatment remain significant issues due to the widespread use of PSA screening, which often detects indolent cancers that may not require intervention, subjecting patients to unnecessary treatments and associated side effects [3,8]. While active surveillance (AS) is increasingly used for low-risk cases, the lack of consensus on inclusion criteria, optimal follow-up schedules, and clear definitions of disease progression limits its effectiveness and broader application [3,8]. Furthermore, treatment side effects are a major concern, with radical prostatectomy and radiation therapy frequently causing long-term complications such as sexual dysfunction (20–70%) and urinary incontinence (15–50%), significantly impacting patients’ quality of life [3]. The benefits of PSA screening in reducing mortality continue to fuel screening controversies, as these must be balanced against the risks of overdiagnosis and overtreatment, creating an ongoing debate in clinical practice [2]. Additionally, disparities in care highlight the stark differences in prostate cancer incidence and mortality between developed and developing countries, driven by variations in access to healthcare, screening availability, and lifestyle factors [2,7]. Finally, emerging diagnostic techniques, including multiparametric MRI and advanced biomarkers, hold promise for improving detection and risk stratification but remain unstandardized, requiring further refinement to integrate them effectively into clinical practice and reduce overtreatment while ensuring aggressive disease is identified and managed appropriately [2]. To address these challenges, future research must prioritize improving risk stratification, refining AS protocols, developing targeted therapies, and combining novel diagnostic tools with personalized care strategies to enhance outcomes and minimize unnecessary interventions [2,3,8]. Figure 1 Illustrates future research priorities in detail.

## 3. Importance of Advancements in Diagnosis and Treatment

The importance of advancements in diagnosing and treating PCa cannot be overstated, as these innovations have significantly improved patient outcomes, quality of life, and overall disease management. Recent developments have addressed several key challenges in PCa care, including overdiagnosis, overtreatment, and more precise and personalized approaches [1,9]. One of the most significant advancements in diagnosis has been the introduction of multiparametric magnetic resonance imaging (mpMRI) into the diagnostic pathway. This imaging technique has revolutionized prostate cancer detection by improving the accuracy of clinically significant cancer identification and reducing unnecessary biopsies [1]. The use of mpMRI has led to improved detection of clinically significant prostate cancer, more accurate localization of appropriate biopsy sites, and a reduction in unnecessary biopsies for patients with normal MRI scans [1,10]. Another crucial diagnostic advancement is the development of prostate-specific membrane antigen (PSMA) positron emission tomography/computed tomography (PET/CT). This imaging modality has shown superior accuracy in staging PCa compared to conventional imaging methods. Australian-led research has demonstrated that PSMA PET/CT is particularly effective in detecting low-volume metastatic disease, even at low PSA levels [1,11]. In terms of treatment, several significant advancements have emerged as follows:

Prostate cancer treatment has seen several significant advancements, transforming the management landscape. Active surveillance (AS) has become an established strategy for low-risk, localized prostate cancer, aimed at avoiding or delaying unnecessary treatment and associated side effects. It involves close monitoring through regular PSA tests, digital rectal examinations, and periodic biopsies, with the goal of curative intervention if cancer progresses AS is typically recommended for patients with PSA < 10 ng/mL, Gleason score 6, and clinical stage T1c or T2a. Distinct from watchful waiting, AS targets curative outcomes rather than symptom management. Studies confirm its safety and similar long-term survival rates compared to immediate treatment, reducing overtreatment while maintaining efficacy [1,12]. Focal therapy, a minimally invasive option for carefully selected patients, provides localized treatment with reduced treatment-related morbidity [13]. Common focal therapy modalities include high-intensity focused ultrasound (HIFU), which uses focused sound waves to ablate targeted prostate tissue, and cryoablation, which destroys cancer cells through cycles of freezing and thawing [14]. These approaches aim to eradicate clinically significant lesions while preserving urinary and sexual function, and are increasingly considered for men with low-to-intermediate risk, localized prostate cancer [15]. Emerging evidence suggests that focal therapies can achieve good oncologic control with fewer side effects compared to whole-gland treatments, though long-term comparative data are still needed [16].

For metastatic castration-resistant prostate cancer (mCRPC), novel hormone therapies such as Abiraterone and Enzalutamide have significantly improved outcomes by targeting the androgen receptor (AR) signaling pathway, a critical driver of prostate cancer progression [17]. Additionally, immunotherapy has advanced with Sipuleucel-T, an autologous active cellular immunotherapy for mCRPC, mobilizing the immune system to target and destroy cancer cells, with potential long-term efficacy [18]. Precision medicine has brought revolutionary progress with PARP inhibitors, particularly for patients harboring specific genetic mutations like BRCA1/2. These drugs, including Olaparib and Rucaparib, inhibit the PARP enzyme in DNA repair, inducing cancer cell death in tumors with defective DNA repair pathways. Molecular profiling helps identify candidates for these therapies, which have shown improved progression-free and overall survival [19,20,21,22,23]. Current studies aim to identify additional genetic markers for PARP response and explore combination strategies to enhance efficacy and overcome resistance. Lastly, combination therapies have emerged as a promising approach for metastatic disease, targeting multiple oncogenic pathways simultaneously to improve outcomes and reduce resistance [24]. These therapeutic advancements underscore the transition toward more personalized and precise treatment strategies, addressing the complexity of prostate cancer biology while minimizing side effects and improving survival (Figure 2).

### 3.1. Metastatic Castration-Sensitive Prostate Cancer (mCSPC)

In recent years, the treatment landscape for mCSPC has evolved significantly. While Androgen Deprivation Therapy (ADT) has long been the cornerstone of treatment, combination therapies have become the new standard of care, offering improved survival outcomes [25].

### 3.2. Doublet Therapies

Several landmark phase 3 trials have demonstrated that adding either chemotherapy (Docetaxel) or androgen receptor pathway inhibitors (ARPIs) to ADT significantly enhances overall survival, particularly for patients with high-volume, high-risk, or de novo metastatic disease [25]. These combinations, known as doublet therapies, include ADT plus any of these chemicals: Docetaxel [26], Abiraterone acetate/Prednisone [27], Apalutamide [28], Enzalutamide [29], or Darolutamide [30].

### 3.3. Triplet Therapies

More recently, triplet therapies have shown promise in extending survival even further. The most compelling evidence supports the use of ADT with Abiraterone and Docetaxel or ADT with Darolutamide and Docetaxel. These combinations have demonstrated improved overall survival compared to ADT plus Docetaxel alone [31,32].

### 3.4. Metastatic Castration-Resistant Prostate Cancer (mCRPC)

For patients with mCRPC, Docetaxel-based combination therapies have been extensively studied. Clinical trials have combined various agents with Docetaxel, including Tyrosine kinase inhibitors, Antiangiogenic agents, Bone-targeted agents, BCL-2 inhibitors, Immunologic agents, and Vitamin D analogs [26]. While some combinations have shown promising results in phase II studies, others have led to increased toxicity or failed to improve survival in phase III trials [26].

### 3.5. Radionuclide Therapy Combinations

Radium-223, a radiopharmaceutical, has shown efficacy in mCRPC patients with bone metastases. Recent studies have explored combining radium-223 with Enzalutamide. This combination improved progression-free survival and treatment completion rates in patients with mCRPC and bone metastases [33]. Furthermore, combinations of radionuclide therapy, hormone therapy, chemotherapy, and immunotherapy are being investigated to enhance the efficacy of radionuclide treatments [34].

Despite the advancements in combination therapies, several challenges remain, such as:Optimizing therapy sequences to balance disease control and treatment burdens.Identifying clinical and biological subgroups that could benefit from personalized treatment strategies.Narrowing the gap between evidence-based guidelines and real-world practice, as many patients still do not receive recommended combination therapies [25].

Although current research is focused on refining treatment approaches and utilizing biomarkers to guide therapy selection, aiming to offer more personalized and adaptive strategies for prostate cancer management [25], combination therapies have significantly improved outcomes for patients with metastatic prostate cancer. As research continues, the Uro-oncology community must align clinical practices with evolving evidence to deliver the most effective care for this incurable disease. These advancements have collectively improved patient outcomes, with men living considerably longer even with metastatic castration-resistant disease [35]. Moreover, the focus on reducing overdiagnosis and overtreatment has helped minimize unnecessary interventions and their associated side effects, improving the quality of life for many patients. However, it is important to note that despite these significant advancements, prostate cancer remains a complex and heterogeneous disease. Ongoing research is crucial to refine diagnostic techniques further, develop more effective treatments, and identify predictive biomarkers to guide personalized treatment decisions [35].

In conclusion, these advancements are important because they can provide more accurate diagnoses, offer tailored treatment options, improve survival outcomes, and enhance the quality of life for prostate cancer patients. As research continues, we can expect further innovations that will continue transforming the landscape of prostate cancer management.

### 3.6. Long-Term Impacts of PCa Advanced Therapies

The long-term impacts of advanced therapies for prostate cancer have significant implications for patient survival, quality of life, and healthcare costs. Advanced therapies have shown promising results in extending survival for patients with metastatic prostate cancer. For instance, in the European Randomized Study of Screening for Prostate Cancer (ERSPC) trial, PSA screening was associated with a 21% relative reduction in prostate cancer mortality after 13 years of follow-up [36]. As detailed earlier, novel treatments like abiraterone, enzalutamide, and apalutamide have demonstrated survival benefits when added to androgen deprivation therapy (ADT) for metastatic hormone-sensitive prostate cancer [37]. In addition, the ProtecT trial found that at a 10-year follow-up, prostate cancer survival was approximately 99% among men with screen-detected prostate cancer across radical prostatectomy, radiation therapy, and active surveillance arms [36]. However, the impact of long-term survival remains unclear in some cases. While early active treatment for screen-detected prostate cancer may reduce the risk of metastatic disease, its long-term implications for prostate cancer mortality is still uncertain [36].

Furthermore, advanced therapies can positively and negatively affect patients’ quality of life. Treatments like radical prostatectomy and radiation therapy are associated with side effects that can significantly impact quality of life, including urinary incontinence, erectile dysfunction, and bowel problems [38]. In the ProtecT quality of life study, radical prostatectomy was associated with worse outcomes in erectile dysfunction and urinary incontinence compared to radiotherapy and ADT [38]. However, no significant differences were observed among treatment groups in general health-related quality of life, cancer-related quality of life, depression, or anxiety [38].

The economic burden of advanced prostate cancer treatments is another substantial factor. A study found that the mean annualized direct medical cost per patient with metastatic castration-resistant prostate cancer was USD 7060, with 91% related to medication costs [39]. Another study reported that the average total cost of treatment for prostate cancer with bone metastases was EUR 13,051 per patient over a 24-month follow-up period [40]. A cost-effectiveness analysis found that adding abiraterone to ADT resulted in an incremental cost-effectiveness ratio of EUR 39,814 per quality-adjusted life-year gained compared to ADT plus docetaxel [37]. Chiefly, the long-term impacts of advanced therapies highlight several key points. Firstly, survival benefits must be weighed against quality-of-life considerations and treatment-related side effects. Secondly, the high costs of newer therapies raise questions about healthcare resource allocation and the need for cost-effective treatment strategies. There is also a need for longer-term follow-up studies to fully understand the impacts of these therapies on survival and quality of life over extended periods. Finally, personalized treatment approaches may become increasingly important to optimize outcomes while managing costs and side effects for individual patients. These long-term impacts underscore the complexity of managing advanced prostate cancer and the need for ongoing research to improve patient outcomes while addressing the economic challenges posed by these advanced therapies [41,42].

## 4. Advances in Prostate Cancer Diagnostics

Advances in diagnostic techniques for prostate cancer have significantly improved the ability to detect and characterize the disease, particularly through biomarker-based and imaging diagnostics. These new approaches aim to enhance the accuracy of prostate cancer detection while reducing unnecessary biopsies and overtreatment.

### 4.1. A Biomarker-Based Approach

The Prostate Health Index (PHI), 4K Score, and Michigan Prostate Score (MiPS) are three notable biomarker-based diagnostic tools.

#### 4.1.1. Prostate Health Index (PHI)

The Prostate Health Index (PHI) is a promising blood test that combines total PSA, free PSA, and [-2] proPSA into a single score. PHI has demonstrated superior performance in distinguishing prostate cancer on biopsy compared to PSA or percentage-free PSA alone. PHI showed greater specificity for detecting prostate cancer when sensitivity was set at 80–95% and approved for men aged 50 and older with PSA levels between 4 and 10 ng/mL and normal digital rectal examination results [43,44]. PHI has demonstrated superior performance compared to traditional PSA testing. It provides higher specificity for detecting clinically significant prostate cancer, the ability to reduce unnecessary biopsies by up to 30.1% at 90% sensitivity, an Area Under the Curve (AUC) of 0.708 for overall prostate cancer detection, and outperforming individual PSA components [44]. PHI has shown promise in improving prostate cancer detection. In a study of 658 men with PSA 4–10 ng/mL, PHI demonstrated an AUC of 0.698 for detecting clinically significant prostate cancer, outperforming PSA alone (AUC 0.549) [45]. A PHI cutoff of 28.6 could have avoided 30% of unnecessary biopsies while maintaining 90% sensitivity for detecting clinically significant cancer [45].

#### 4.1.2. The 4K Score

The 4K Score is an advanced blood-based diagnostic test designed to improve risk stratification for aggressive prostate cancer. It combines measurements of four Kallikrein markers—total PSA, free PSA, intact PSA, and human Kallikrein 2 (hK2)—along with clinical information such as patient age, digital rectal examination results, and prior biopsy history. This comprehensive approach provides a highly accurate risk assessment for detecting high-grade prostate cancer, defined as a Gleason score ≥ 7 [46,47]. Integrating biochemical and clinical factors enables the 4K Score to achieve a high level of precision, with a pooled Area Under the Curve (AUC) of 0.81 across multiple studies for identifying high-grade disease [47]. Importantly, this test has demonstrated the potential to reduce unnecessary biopsies by 30–58%, depending on the selected risk threshold, without compromising the detection of clinically significant cancers. This reduction in invasive procedures minimizes patient discomfort and the risks associated with unnecessary interventions, such as infection or overtreatment. The 4K Score represents a significant advancement in prostate cancer diagnostics, offering a personalized and evidence-based tool to guide clinical decision-making. By focusing on aggressive cancers, it supports a shift toward more targeted and efficient management strategies, ensuring that patients receive appropriate care while reducing the burden of overtreatment and its associated complications. Future research and continued validation of the 4K Score in diverse populations will further refine its role in optimizing prostate cancer screening and risk assessment [46,47]. The 4K score has improved accuracy in predicting high-grade prostate cancer. In a study of 1012 men, the 4K score showed an AUC of 0.82 for predicting high-grade cancer (Gleason ≥ 7), compared to 0.74 for PSA alone (*p* < 0.001) [48]. A 4Kscore cutoff of 7.5% could have avoided 43% of biopsies while missing only 2.4% of high-grade cancers [48].

#### 4.1.3. Michigan Prostate Score (MiPS)

The Michigan Prostate Score (MiPS) is a urine-based test that combines PSA with two urinary biomarkers for prostate cancer—urinary Prostate cancer antigen 3 (PCA3) mRNA and urinary TMPRSS2: ERG gene fusion. MiPS improved prostate cancer detection, including more aggressive forms, compared to traditional PSA-based models [49]. The test provides men with a more individualized risk assessment for prostate cancer [50]. These biomarker-based diagnostic techniques represent significant advancements in prostate cancer detection and risk stratification. By offering more accurate and personalized risk assessments, these tests have the potential to reduce unnecessary biopsies, decrease overdiagnosis, and improve patient outcomes through more targeted interventions [50]. Figure 3 shows advances in prostate cancer diagnosis, highlighting the role of mpMRI, biomarkers, and PSMA-PET/CT.

### 4.2. B Imaging

Recent imaging advancements have significantly improved the diagnosis, staging, and management of prostate cancer. Three key technologies have emerged as particularly promising: Multiparametric Magnetic Resonance Imaging (mpMRI), Prostate-Specific Membrane Antigen (PSMA) PET Imaging, and MRI–Ultrasound Fusion-Guided Biopsies.

#### 4.2.1. Multiparametric Magnetic Resonance Imaging (mpMRI)

Multiparametric MRI has revolutionized prostate cancer detection and risk classification. This technique combines T2-weighted, diffusion-weighted, and dynamic contrast-enhanced imaging to provide detailed anatomical and functional information about the prostate [51]. Key benefits of mpMRI include high sensitivity (96%) and negative predictive value (92%) for detecting significant prostate cancer [51], the ability to differentiate between benign and potentially malignant conditions [45,52], and improved detection of clinically significant cancers while reducing the detection of insignificant cancers [51]. mpMRI is now recommended as the first-line investigation for suspected clinically localized prostate cancer [52]. It has also shown promise in guiding biopsies, staging, and monitoring patients undergoing active surveillance. In a study of 576 men, mpMRI demonstrated a sensitivity of 93% and specificity of 41% for detecting clinically significant prostate cancer (Gleason score ≥ 7) [53]. The addition of mpMRI to standard biopsy increased the detection of clinically significant cancer by 13% (95% CI: 8–19%, *p* < 0.001) [53].

#### 4.2.2. Prostate-Specific Membrane Antigen (PSMA) PET Imaging

Prostate-Specific Membrane Antigen (PSMA) PET imaging has emerged as a powerful tool for prostate cancer diagnosis and management. This molecular imaging technique uses radioligands that target PSMA, which is overexpressed in most prostate cancer cells [54]. PSMA PET imaging offers several advantages, such as superior accuracy (92%) compared to conventional imaging (65%) for staging high-risk prostate cancer [54]; high sensitivity in detecting metastases, even in patients with low PSA levels [54]; and the ability to guide treatment decisions, especially in cases of biochemical recurrence [55]. Recent studies have shown that PSMA PET can improve detection rates with rising PSA levels, from 33% for PSA levels below 0.19 ng/mL to 95% for PSA levels > 2.0 ng/mL [56]. This technique has been incorporated into several guidelines for prostate cancer management, including those of the American Urological Association and the European Association of Urology [54]. To show the PSMA PET/CT superior performance for staging prostate cancer, a meta-analysis found PSMA PET/CT had a pooled sensitivity of 77% (95% CI: 71–82%) and specificity of 97% (95% CI: 95–98%) for detecting lymph node metastases [57]. For primary tumor detection, PSMA PET/CT demonstrated a sensitivity of 74% (95% CI: 67–80%) and specificity of 96% (95% CI: 85–99%) [57].

#### 4.2.3. MRI–Ultrasound Fusion-Guided Biopsies

MRI–Ultrasound fusion-guided biopsies combine the detailed imaging of mpMRI with real-time ultrasound guidance to improve the accuracy of prostate biopsies [58]. This technique offers several benefits, such as improved detection of clinically significant cancers compared to standard biopsy techniques (74); reduced detection of low-risk, clinically insignificant cancers (74); and the ability to target specific areas of concern identified on MRI [58]. A study of 1003 men undergoing both targeted and standard biopsies found that MRI–Ultrasound fusion-guided biopsies diagnosed 30% more high-risk cancers and 17% fewer low-risk cancers compared to standard biopsies [58]. These imaging advancements have significantly improved the ability to detect, stage, and manage PCa. They offer more accurate diagnosis, better risk stratification, and the potential for more personalized treatment approaches. As these technologies continue to evolve and become more widely available, they will likely play an increasingly important role in prostate cancer care.

## 5. Artificial Intelligence and Machine Learning in PCa

Recent advances in artificial intelligence (AI) and machine learning (ML) are transforming prostate cancer diagnosis and treatment. AI-powered tools now improve the accuracy of imaging interpretation, such as MRI and pathology slides; help identify high-risk tumors; and support more precise, personalized treatment decisions. For example, AI can create detailed 3D maps of prostate tumors to guide focal therapies and predict treatment success. AI-based risk models also aid in post-surgical management by predicting recurrence and informing follow-up care. While these technologies offer significant promise for enhancing outcomes, challenges remain in data quality, validation, and clinical integration (Figure 4).

AI has become integral to prostate imaging, especially mpMRI and PSMA-PET/CT (Figure 5). Convolutional neural networks (CNNs) now match expert radiologists in detecting and classifying lesions. In a study of over 1000 patients, an AI system achieved an AUC of 0.84 for detecting clinically significant prostate cancer (csPCa), comparable to experts. AI-assisted mpMRI improves lesion detection, reduces inter-reader variability, and enhances biopsy targeting. Incorporating MRI into risk calculators increases PCa prediction accuracy, raising AUC from 64% to 84% and reducing false-positives, thus avoiding unnecessary biopsies. In PSMA-PET/CT, AI automates lesion detection, tumor quantification, and staging [59,60,61], improving treatment planning and response monitoring in both early and advanced PCa.

### 5.1. Predictive Models and Risk Stratification

AI and ML enable predictive models that combine clinical, imaging, and genetic data for better risk stratification. For example, an AI risk model integrating clinical factors, polygenic scores, and MRI achieved an AUC of 0.84 for csPCa prediction [62]. AI-augmented nomograms outperform traditional calculators, with one reporting an AUC of 0.89 for predicting recurrence after prostatectomy [63]. Deep learning also improves genomic risk prediction (AUC 0.79) [64]. Novel blood, urine, and genetic biomarkers further enhance AI’s role in risk assessment [65].

### 5.2. AI in Pathology and Histopathology

AI advances pathology by improving Gleason grading accuracy and consistency, reducing variability, and speeding diagnosis. Validated AI systems as second-read tools have shown improved accuracy over three years, especially for borderline or complex cases. AI also facilitates tumor characterization, aiding precise treatment decisions.

### 5.3. Precision Medicine Applications

AI supports precision medicine in PCa by enabling personalized treatment strategies. Models predict treatment response and recurrence risk using imaging, genetic, and clinical data. AI informs targeted therapy and active surveillance decisions, minimizing overtreatment [62,63,66,67,68].

### 5.4. Predictive Analytics and Outcome Prediction

AI models predict outcomes such as risk, metastasis, and mortality, supporting clinical decision-making. Their adoption depends on integration into EMRs and regulatory approval.

### 5.5. Natural Language Processing (NLP) for Clinical Data Extraction

NLP extracts critical information from EHRs for PCa management, outperforming traditional coding in identifying metastasis. It aids in risk stratification and staging, and automates data extraction, though challenges remain with ambiguous language.

## 6. Challenges and Future Opportunities in AI for Prostate Cancer

Despite advances, clinical AI implementation in prostate cancer (PCa) faces challenges in data quality, validation, ethics, and regulation.

### 6.1. Challenges in AI Implementation

#### 6.1.1. Data Quality and Bias

AI models require diverse, high-quality datasets; underrepresentation can cause biased outcomes and inequitable care.

#### 6.1.2. Validation and Generalization

AI often lacks generalizability across settings. The PANDA challenge showed models may not perform well outside specific environments. External, multi-institutional validation is needed, as are standardized metrics. Integration into EMRs, ethical concerns, and data governance remain hurdles [69,70,71,72].

#### 6.1.3. Regulatory and Standardization Issues

Lack of standardized guidelines and regulatory frameworks limits adoption. High costs and workflow complexity further hinder routine use [52]. Small, non-representative datasets and complex algorithms impede interpretability and clinician trust [73,74].

## 7. Future Opportunities

AI offers significant future opportunities in prostate cancer care by enhancing diagnostic accuracy and consistency in imaging interpretation, such as mpMRI and PSMA-PET/CT, and supporting risk stratification and personalized treatment recommendations. AI streamlines radiotherapy planning by automating tasks like organ-at-risk contouring, reducing planning time, and increasing precision. Integrating multimodal data—including clinical, imaging, genetic, and pathology information—enables comprehensive risk assessment and more individualized care, with combined models showing improved prediction of disease progression. Additionally, AI acts as a valuable digital assistant for pathologists and radiologists, improving workflow efficiency and diagnostic accuracy in tasks such as Gleason grading and lesion detection [75,76].

Artificial Intelligence (AI) and Machine Learning (ML) are powerful tools in prostate cancer (PCa) diagnosis and risk stratification, improving accuracy, efficiency, and patient outcomes. AI enhances imaging interpretation and predictive models, especially in MRI and PET/CT. AI algorithms, particularly convolutional neural networks (CNNs), match experienced radiologists in detecting and classifying suspicious lesions on mpMRI, improving lesion detection, reducing inter-reader variability, and aiding biopsy planning [77]. Studies show AI can standardize lesion detection, with an AI system achieving an AUC of 0.84 for clinically significant PCa, comparable to experts. In PSMA PET/CT, AI enables automatic lesion detection and classification, tumor quantification, and prognostication, with performance similar to human interpretation and added insights. Figure 6 illustrates AI integration in mpMRI and PSMA-PET/CT for PCa diagnosis.

## 8. Predictive Models for Risk Stratification in Prostate Cancer

AI and ML are advancing risk stratification in prostate cancer by combining clinical, genetic, and imaging data to identify high-risk patients more accurately. Recent models integrating factors such as age, family history, PSA, polygenic risk scores, and MRI findings have shown strong predictive performance, with one example achieving a hazard ratio of 3.058 per standard deviation of 5-year risk and a Harrell’s C-index of 0.811 in validation [78]. Table 1 summarizes AI and ML applications in imaging, predictive modeling, and treatment planning for prostate cancer.

Recent advancements in AI and ML applications in PCa management area include:Comprehensive Risk Models: for example, a study by Seibert et al. developed a model combining clinical factors, polygenic risk scores, and MRI findings, achieving an Area under the curve (AUC) of 0.84 for predicting csPCa [84].AI-Enhanced Nomograms: Researchers have created AI-augmented nomograms that outperform traditional risk calculators in predicting PCa outcomes. One model achieved an AUC of 0.89 for predicting biochemical recurrence after radical prostatectomy [85].Deep Learning for Genomic Risk Prediction: a deep learning model analyzing genomic data demonstrated superior performance in predicting PCa risk compared to traditional polygenic risk scores, with an AUC of 0.79 [86].

### 8.1. Clinical Validation Gaps: ProFound AI vs. Radiologists

Although ProFound AI is FDA-cleared for prostate cancer procedure planning and has demonstrated improvements in workflow efficiency and precision, direct comparative data on sensitivity and specificity versus radiologists in prostate biopsy guidance remain limited (https://www.massdevice.com/profound-medical-fda-clearance-ai-prostate-cancer,/, accessed 20 May 2025). In breast cancer screening, ProFound AI has shown high sensitivity (up to 92%) but still does not consistently outperform experienced radiologists in all settings, and most published validation studies are in breast imaging rather than prostate applications [87]. Importantly, many FDA-cleared AI tools, including ProFound AI, have limited prospective clinical validation in prostate cancer, highlighting the need for randomized trials comparing AI-guided and radiologist-led biopsy strategies to establish real-world effectiveness.

### 8.2. MRI-Based Prediction Models

Incorporating MRI findings into risk calculators has further improved the accuracy of prostate cancer prediction. These models have shown the potential to reduce unnecessary biopsies while maintaining a high detection rate for csPCa [88]. There are several key findings for this model. For instance, when adding MRI parameters to baseline clinical models, the area under the curve (AUC) improves from 64% to 84%. Additionally, potential reduction is observed in false-positive rates (46% vs. 92%), with only a small decrease in true-positive rates (89% vs. 99%) at a 20% risk threshold. Avoiding 18 out of 100 unnecessary biopsies is possible without missing clinically significant cancers [88]. These findings demonstrate that integrating MRI data into risk prediction models not only enhances diagnostic performance but also has practical implications for clinical decision-making. By reducing unnecessary biopsies—without compromising the detection of clinically significant prostate cancer—these tools help minimize patient burden, procedural risks, and healthcare costs. However, widespread clinical adoption depends on the availability of standardized imaging protocols, access to high-quality MRI interpretation, and validation across diverse populations. While promising, these models should be implemented with consideration of real-world constraints and the need for prospective trials to confirm their utility in routine practice.

### 8.3. Novel Biomarkers and Risk Calculators

The integration of novel biomarkers and risk calculators has further enhanced risk stratification. Blood-based biomarkers like the Prostate Health Index (PHI) and 4Kscore show similar ability to predict csPC. Urinary biomarkers such as PCA3, when combined with other markers like TMPRSS2-ERG, improve predictive performance. New risk models incorporating genetic markers, such as SelectMDx and Stockholm-3 model (S3M), promise to improve risk assessment [89]. While AI and ML show great promise in prostate cancer diagnosis and risk stratification, many AI models lack external validation and prospective design. Large-scale, multi-institutional studies are needed to ensure robustness and generalizability [69,81]. There is a need for standardized reporting of performance metrics and consistent methodologies across studies to allow for meaningful comparisons [69]. Hence, developing user-friendly interfaces and seamless integration into clinical pathways is crucial for widespread adoption [90]. Importantly, addressing issues related to data privacy, algorithmic bias, and transparency in AI decision-making processes is essential [88,91]. Ultimately, prospective clinical trials will be necessary to establish the benefit of AI in patient management and outcomes [69]. In conclusion, AI and ML are poised to revolutionize prostate cancer diagnosis and risk stratification. By enhancing imaging interpretation and developing sophisticated predictive models, these technologies offer the potential for more personalized and accurate patient care. However, continued research, validation, and careful implementation will be crucial to realizing their full potential in clinical practice. Together, these advancements in biomarker discovery and AI-driven risk stratification represent a significant step toward more precise and individualized prostate cancer management. While tools like PHI, 4Kscore, and SelectMDx have improved pre-biopsy risk assessment, their impact on clinical practice remains contingent on widespread validation and ease of integration into routine workflows. The growing application of AI and machine learning in this space holds great promise but is currently tempered by limitations such as insufficient external validation, inconsistent reporting standards, and the need for regulatory oversight to address issues of transparency and bias. For these technologies to translate into real-world benefit, they must be embedded into patient-centered care models, supported by prospective trials, and made accessible across diverse healthcare settings. Until then, their adoption should be guided by both evidence and caution.

### 8.4. Precision Medicine Applications

Integrating AI in PCa management is paving the way for more personalized treatment approaches. In treatment response prediction, AI models analyzing pre-treatment MRI and clinical data can predict response to specific therapies, allowing for more informed decision-making [67]. In addition, in recurrence risk assessment, machine learning algorithms that integrate post-treatment imaging and molecular markers have shown promise in predicting the risk of disease recurrence [92]. Furthermore, in adaptive therapy planning, AI systems can continuously learn from patient outcomes to refine treatment protocols, potentially leading to adaptive, personalized therapy regimens [66].

In conclusion, AI and ML are transforming PCa diagnosis and management by enhancing imaging interpretation, improving risk stratification, and enabling more precise, personalized treatment strategies. As these technologies continue to evolve and integrate with clinical practice, they promise to significantly improve patient outcomes in prostate cancer care.

### 8.5. Advances in Prostate Cancer Treatment

The landscape of PCa treatment has undergone significant transformations in recent years, with numerous advancements to improve patient outcomes and quality of life. These innovations span various treatment modalities, offering new hope for patients at different stages of the disease. One notable advancement is in the field of radiation therapy. Intensity-modulated radiation therapy (IMRT) and proton beam radiation have emerged as more precise methods of delivering radiation to tumor sites while minimizing damage to surrounding healthy tissues [93]. These techniques allow higher radiation doses to be administered with reduced side effects, potentially improving treatment efficacy. Image-guided approaches have further enhanced the precision of radiation delivery, ensuring that the treatment targets the tumor accurately even as the prostate’s position may shift slightly from day to day [93]. Surgical techniques have also seen significant improvements. The adoption of robotic-assisted laparoscopic prostatectomy has led to more precise surgeries with potentially faster recovery times and fewer complications [94]. This minimally invasive approach allows surgeons to operate with enhanced visualization and dexterity, potentially improving outcomes for patients undergoing surgical intervention. Due to the improvement of systemic treatments, the development of novel hormone therapies has expanded the options for patients with advanced PCa. These therapies, which target the androgen receptor signaling pathway more effectively than traditional androgen deprivation therapy, have shown promise in extending survival and improving the quality of life for patients with metastatic disease [95,96]. Recent approvals include Abiraterone Acetate, Enzalutamide, and Apalutamide, which have demonstrated significant benefits in various stages of prostate cancer [95].

Immunotherapy approaches, such as Sipuleucel-T, have introduced the concept of harnessing the patient’s immune system to fight PCa [95]. Sipuleucel-T was the first immunotherapy product and the first therapeutic vaccine for any cancer approved by the US FDA [97]. While the benefits of immunotherapy in PCa have been modest compared to some other cancer types, ongoing research continues to explore new ways to enhance immune responses against prostate tumors. For instance, chimeric antigen receptor (CAR) T-cell therapy is being investigated in clinical trials, showing encouraging early results [93].

Precision medicine has also made significant advancements in PCa treatment. Genetic testing and molecular profiling of tumors are increasingly used to guide treatment decisions, allowing for more personalized approaches based on the specific characteristics of each patient’s cancer [95,98]. This approach has led to the development of targeted therapies, such as PARP inhibitors for patients with specific genetic mutations [96]. The rapid evolution of PCa treatment options has significantly improved outcomes for many patients. As research continues, we can expect further refinements in existing therapies and the emergence of new treatment modalities, offering hope for even better management of PCa in the future [99]. Innovations such as targeted therapies like PSMA-directed radiopharmaceuticals and novel hormone therapies offer new hope for patients with advanced PCa, potentially extending survival and improving quality of life. Nanomedicine or using nanoparticle-based therapies have emerged as promising approaches for the treatment of PCa, offering potential advantages over conventional treatments. Nanocarriers have revolutionized drug delivery in prostate cancer treatment by addressing limitations of traditional chemotherapy, such as poor drug solubility, rapid clearance, and systemic toxicity [100,101]. Several types of nanocarriers have been developed for PCa treatment, including Liposomes, Polymeric Nanoparticles, Gold Nanoparticles, and Silica-Based Nanoparticles [100,102]. As noted earlier, PSMA is one of the most distinctive and highly selective biomarkers for PCa. PSMA-targeted nanomedicines have shown promise in enhancing treatment efficacy and reducing adverse effects [103]. These nanoparticles can be potentially used for targeted drug delivery, imaging and diagnostics, and combination therapies, for example, in chemotherapy and radiotherapy [103]. While nanoparticle-based therapies show great promise, several challenges remain. For instance, the pharmacokinetics and biodistribution of these new chemicals should be optimized, and their production for clinical use needs to be scaled up. Their long-term safety and ability to overcome biological barriers are crucial considerations [100,103]. As research in nanomedicine for PCa continues to progress, further refinements in existing therapies and the emergence of new treatment modalities are expected in the coming years. Figure 7 shows three innovative therapeutic modalities’ efficacy in PCa therapy.

Collectively, these therapeutic advancements signal a profound shift in how prostate cancer is managed—moving from a one-size-fits-all approach to more precise, patient-centered care. Innovations such as image-guided radiotherapy, robotic surgery, next-generation hormone therapies, and PSMA-targeted treatments are not only improving survival metrics but also enhancing quality of life by minimizing side effects and improving functional outcomes. However, several barriers remain. Access to technologies like proton therapy or PSMA radioligands is limited by infrastructure and cost. Similarly, while nanomedicine and immunotherapy hold transformative potential, their integration into clinical practice depends on overcoming challenges related to scale-up, toxicity, and long-term efficacy. As a result, ongoing research, patient stratification strategies, and regulatory pathways will be essential in translating these innovations into widely available and sustainable treatment options. The future of prostate cancer therapy thus lies in combining scientific rigor with equitable and patient-tailored implementation.

### 8.6. Passive vs. Active Targeting in Nanomedicine: Mechanisms, Efficiency, and Clinical Implications

In addition, Nanomedicine utilizes both passive and active targeting strategies to improve drug delivery. Passive targeting relies on the enhanced permeability and retention (EPR) effect, where nanoparticles accumulate in tumors due to leaky vasculature and impaired lymphatic drainage, but typically only about 1% of the injected dose reaches the tumor, with most accumulating in healthy organs like the liver and spleen [104,105]. Active targeting uses ligands (such as antibodies or peptides) on the nanoparticle surface to bind specific receptors overexpressed on tumor cells, increasing selectivity and cellular uptake, though no active-targeted nanomedicines are yet FDA-approved due to biological and clinical challenges [105,106]. Combining both approaches may further enhance therapeutic efficacy in prostate cancer and other solid tumors.

### 8.7. Advances in Radiation Therapy

Radiation therapy for PCa has undergone significant advancements in recent years, improving treatment outcomes and reducing side effects for patients. These developments encompass various aspects of radiotherapy, from imaging techniques to treatment delivery methods. Integrating advanced imaging techniques has revolutionized PCa radiotherapy, marking a crucial step forward in this field. Multiparametric MRI and MRI–ultrasound fusion-guided prostate biopsy are two of these advancements that have enhanced the detection of clinically significant PCa and provided crucial information about intraprostatic lesions [107]. These imaging advancements allow for more precise target volume definition and treatment planning [108]. Another imaging technique, Intensity-Modulated Radiation Therapy (IMRT), allows for dose escalation without significantly increasing treatment-related morbidity [108]. IMRT’s ability to conform radiation doses to complex target volumes while sparing surrounding healthy tissues has improved the therapeutic ratio [109]. Image-guided radiation Therapy (IGRT), also considered a recent advanced radiation that uses X-rays and scans to improve the accuracy of radiation therapy [110], has enhanced the precision of radiation delivery. This technology ensures that the radiation beam accurately targets the prostate, even with slight movements or changes in organ position [107,108]. In terms of safe and effective radiation, recent studies have explored hypofractionation, which involves delivering larger doses of radiation over fewer treatment sessions. Moderate and extreme hypofractionation have shown promise in improving patient convenience without compromising treatment efficacy [107,109]. However, Stereotactic Body Radiotherapy (SBRT), a form of extreme hypofractionation, has emerged as a potential option for select patients, offering shorter treatment courses with comparable outcomes [111]. Among all these techniques, proton beam therapy has gained attention for its potential to spare normal tissues further while maintaining target coverage. It is a non-invasive radiation treatment that targets and destroys PCa cells using proton particles [112]. Proton beams deliver a precise radiation dose to the tumor while minimizing exposure to surrounding organs at risk [107,111]. MRI-guided radiotherapy (MRIgRT) is another PCa treatment that uses MRI to improve the accuracy of radiation delivery. This system combines MRI scanners with linear accelerators and represents a significant advancement. These hybrid machines offer superior image quality and enable daily adaptive replanning, potentially allowing for dose escalation in challenging cases and more selective sparing of healthy tissues [113,114] (Figure 8).

### 8.8. Combination Therapies in PCa: Androgen Deprivation Therapy (ADT)

The integration of Androgen Deprivation Therapy (ADT) with definitive radiotherapy has led to improved outcomes for certain subgroups of patients, particularly those with high-risk disease [108]. Ongoing research is exploring optimal combinations and durations of ADT with various radiotherapy techniques [115]. Researchers are investigating the potential benefits of combining radiotherapy with new systemic therapies, including next-generation androgen receptor-targeted agents. These combinations hold promise for improving outcomes in patients with locally advanced PCa [108,115]. The evolving use of ADT in combination with radiotherapy underscores a strategic shift toward maximizing therapeutic synergy in high-risk and locally advanced prostate cancer. Evidence supporting these combinations has already influenced clinical guidelines, particularly in defining treatment approaches for patients with adverse features. However, the heterogeneity of patient response—along with uncertainties around the optimal duration and sequencing of ADT—remains a critical area for refinement. The integration of next-generation androgen receptor inhibitors may further enhance therapeutic efficacy, but long-term safety, cost, and patient selection criteria need to be clearly established through prospective trials. As we refine these combination strategies, balancing efficacy with tolerability will be key to improving both survival and quality of life.

### 8.9. Emerging Frontiers: Radionuclide Therapy and Oligometastatic Disease

Recent advances in radionuclide therapy, particularly with beta-emitting 177Lu-PSMA, have shown promising results for patients with metastatic castration-resistant PCa. Ongoing research explores alpha-emitting radionuclides and combination strategies to combat treatment resistance [116]. Exciting breakthroughs are revealing potential new indications for radiotherapy in oligometastatic PCa. SBRT and other advanced techniques are being studied for their efficacy in treating limited metastatic disease [115]. In conclusion, the field of radiation therapy for PCa continues to evolve rapidly. These advancements aim to improve tumor control, reduce treatment-related toxicities, and enhance the overall quality of life for patients with PCa. As research progresses, even more refined and personalized approaches to PCa radiotherapy will likely emerge, further improving patient outcomes across all risk groups. Table 2 overviews treatment advances in PCa and categorizes surgical, radiation, systemic, and combination therapies with associated outcomes.

### 8.10. FDA-Approved Liquid Biopsy Assays: Guardant360 CDx Utility in CRPC

Recent advances in liquid biopsy have led to the development and FDA approval of comprehensive genomic profiling (CGP) assays, such as Guardant360 CDx, which are now recommended for use in advanced solid tumors, including castration-resistant prostate cancer (CRPC) (references: 1. Guardant360^®^ CDx: Fastest FDA-Approved Liquid CGP Panel, 2. Guardant Complete^®^ for Early and Advanced Stage Cancer, 3. Guardant Health Press Release 2020, 4. FDA Approves Guardant360 CDx as Companion Diagnostic for Osimertinib). Guardant360 CDx is a blood-based next-generation sequencing test that detects actionable mutations (including BRCA1/2 and ATM) in advanced solid tumors, such as castration-resistant prostate cancer (CRPC). It offers a rapid turnaround time (typically <7 days), is covered by Medicare, and is especially valuable when there are insufficient or inaccessible tissue samples. Analytical validation demonstrates high concordance with tissue-based testing, supporting its use for guiding targeted therapies (e.g., olaparib in BRCA-mutated CRPC). Compared to traditional tissue biopsies, Guardant360 CDx enables non-invasive, repeatable genomic profiling, facilitating timely and personalized treatment decisions in advanced prostate cancer [117].

### 8.11. CRISPR-Mediated AR-V7

Recent studies have used CRISPR/Cas9 to investigate and target the AR-V7 splice variant, a key driver of resistance in castration-resistant prostate cancer (CRPC). For example, CRISPR was used to generate CWR22Rv1 cell lines lacking full-length AR but retaining AR-V7, confirming AR-V7’s independent role in tumor progression [118]. Genome-wide CRISPR screens have also identified PRMT1 as a regulator of AR-V7 expression, with PRMT1 inhibition sensitizing cells to antiandrogen therapy [119]. Furthermore, new CRISPR base editing approaches are being explored to directly correct AR-V7 splicing defects, offering future therapeutic potential [119].

**Table 2 ijms-26-05386-t002:** Prostate Cancer Treatment Advances: A Comprehensive Overview.

Category	Treatment	Outcomes	References
Surgical Techniques	Robotic-assisted laparoscopic prostatectomy (RALP)	-Improved precision and control -Less blood loss -Shorter hospital stays -Faster recovery -Similar cancer control rates to open surgery	[25,96]
Radiation Therapy	-Stereotactic Body Radiation Therapy (SBRT)	-High dose delivery in fewer sessions -95.8% five-year survival rate for intermediate-risk localized PC	[9]
	-Intensity-Modulated Radiation Therapy (IMRT) -Proton Beam Therapy -MRI-guided radiotherapy (MRIgRT)	-Improved dose conformity -Reduced toxicity -Better quality of life -Precise dose delivery -Minimized exposure to surrounding organs	[9,25,96,120]
Systemic Treatments	-Abiraterone + ADT -Enzalutamide + ADT -Apalutamide + ADT	-Improved overall survival (HR: 0.63; 95% CI: 0.52-0.76) -Better progression-free survival -Improved overall survival -Reduced risk of radiologic progression or death by 61% -Improved overall survival (HR: 0.67; 95% CI: 0.51-0.89) -Improved radiographic progression-free survival	[96]
	Docetaxel + ADT	-Improved overall survival (57.6 months vs. 47.2 months) -More effective in high-volume disease	[25,96]
	Lutetium-177-PSMA-617	-Improved imaging-based progression-free survival (8.7 vs. 3.4 months) -Improved overall survival (15.3 vs. 11.3 months) in mCRPC	[9]
Combination Therapies	ADT + Radiotherapy	-Improved outcomes for high-risk disease patients	[120]
	ADT + Docetaxel + Abiraterone	-Enhanced metastasis-free survival	[120]
	Radium-223 + Enzalutamide	-33% reduction in radiologic disease progression or death	[99]

### 8.12. Recent Advancements in PCa Clinical Trial Research

The PEACE-3 trial demonstrated that combining Radium-223 with Enzalutamide for first-line metastatic castration-resistant PCa (mCRPC) improved overall survival compared to Enzalutamide alone. The median overall survival (OS) for the combination therapy was 19.0 months vs. 16.0 months for Enzalutamide monotherapy (HR 0.70, 95% CI 0.56–0.88; *p* = 0.002), with a 30% reduction in the risk of death. This combination therapy offers a promising approach for patients with bone metastases, significantly delaying disease progression (median radiographic progression-free survival: 16.2 vs. 11.7 months; HR 0.65, *p* < 0.001) and improving quality of life and overall survival in mCRPC patients [121].

The EMBARK trial led to the FDA approval of Enzalutamide for nonmetastatic PCa with biochemical recurrence at high risk of metastasis. Patients treated with Enzalutamide plus Leuprolide or Enzalutamide alone showed a median metastasis-free survival (MFS) of not reached vs. 25.1 months for Leuprolide monotherapy (HR 0.42, 95% CI 0.30–0.61; *p* < 0.001 for combination; HR 0.63, 95% CI 0.46–0.87; *p* = 0.005 for monotherapy). This expanded treatment option reduces the risk of metastasis or death by 58% (combination) and 37% (monotherapy) for high-risk nonmetastatic PCa. However, grade ≥3 adverse events occurred in 46% (combination), 42% (Enzalutamide alone), and 26% (Leuprolide alone) of patients, highlighting the need to balance efficacy with toxicity and uncertainty about the optimal treatment duration [95,96].

Recent clinical trials, including PEACE-3 and EMBARK, along with the expanded FDA approval of Enzalutamide, mark significant progress in PCa treatment. These advancements offer new strategies for patients across various disease stages, from high-risk nonmetastatic (EMBARK) to metastatic castration-resistant (PEACE-3) PCa. While these developments provide valuable treatment options and opportunities for earlier intervention, they also present challenges such as potential adverse events (e.g., fatigue, cardiovascular risks), high costs, and the risk of overtreatment [122,123]. Ongoing research will be crucial to optimize treatment approaches, identify predictive biomarkers, and further personalize care. These advancements represent important steps towards improving outcomes and quality of life for PCa patients, with the promise of continued refinements in treatment strategies [123] (Figure 9).

## 9. Conclusions and Future Direction

Over the past decade, prostate cancer (PCa) research has progressed from generalized treatment approaches to a deeply personalized paradigm powered by technological and biological innovation. While this transformation has delivered tangible improvements in diagnosis, treatment efficacy, and patient quality of life, the field stands at a pivotal juncture where the integration, validation, and accessibility of these advancements will define the next era of care.

Future research must prioritize clinical translation and real-world validation. Technologies like PSMA-PET, MRI-based predictive models, and AI-powered diagnostic tools have demonstrated superior performance in early trials and retrospective analyses. However, their widespread adoption hinges on multi-institutional validation, standardization of metrics, and seamless integration into clinical workflows. The development of prospective studies evaluating AI-assisted diagnosis and treatment planning—as compared to radiologist-led or clinician-driven decisions—will be essential to determine true clinical value.

Biomarker discovery and validation remain key frontiers. While tools such as PHI, 4Kscore, and SelectMDx have advanced risk stratification, their predictive power varies across populations and clinical contexts. Expanding molecular profiling to include dynamic biomarkers (e.g., AR-V7) and liquid biopsy assays (e.g., Guardant360 CDx) offers a promising avenue to enable longitudinal, non-invasive monitoring and guide adaptive therapeutic strategies.

In treatment, the success of next-generation hormone therapies, PARP inhibitors, immunotherapy, and radioligand approaches opens the door to strategic combinations. However, unresolved questions regarding optimal sequencing, resistance mechanisms, and long-term toxicities require carefully designed clinical trials. Simultaneously, the refinement of focal therapies, radiotherapy dose modulation (e.g., SBRT, MRIgRT), and nanomedicine applications must balance efficacy with cost-effectiveness and equitable access, particularly in resource-limited settings.

Artificial intelligence (AI) and machine learning (ML) represent transformative forces in PCa care. These tools can enhance imaging interpretation, automate pathology, personalize treatment recommendations, and even predict recurrence. Yet, the lack of external validation, concerns about data bias, interpretability, and regulatory pathways continue to limit clinical translation. Future work must emphasize transparency, ethics, and interoperability while designing AI tools that serve as augmentative—not replacement—resources in the clinic.

Finally, there is a need to reimagine prostate cancer management through a patient-centered lens. Precision medicine should not only aim for biological tailoring but also consider patient preferences, psychosocial factors, and survivorship needs. As more patients live longer with PCa, quality-of-life preservation, reduction in overtreatment, and management of long-term toxicities will become increasingly important.

In conclusion, prostate cancer research is experiencing a confluence of innovation spanning diagnostics, therapeutics, and informatics. To fully realize the promise of these advances, the next phase must focus on integration, equity, and evidence-based personalization. With coordinated global efforts in translational research, digital health, and implementation science, the vision of a fully personalized, less invasive, and outcome-driven PCa care model is within reach.

## Figures and Tables

**Figure 1 ijms-26-05386-f001:**
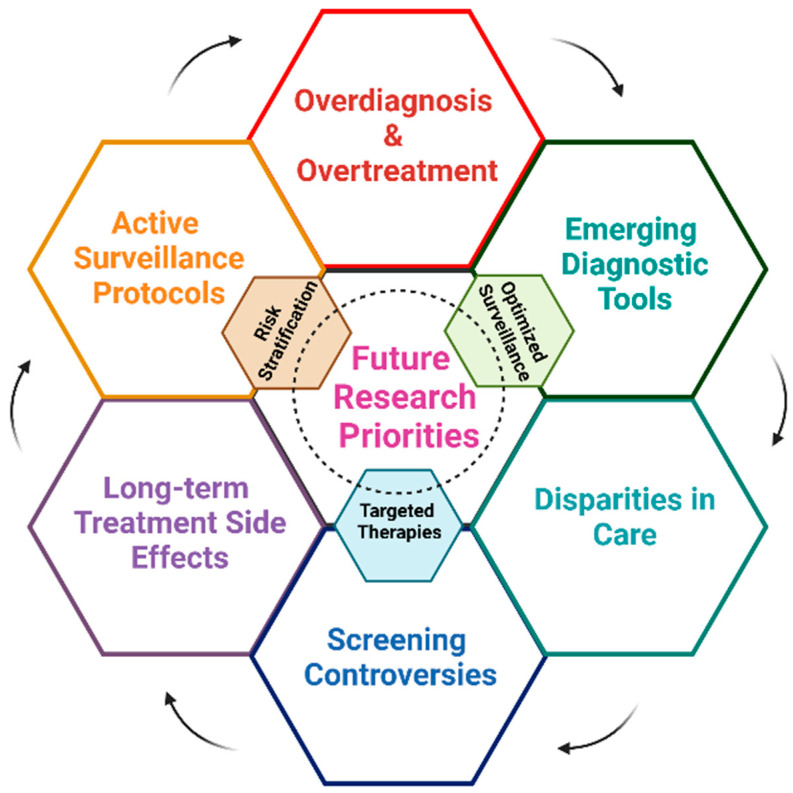
Challenges and Innovation in Prostate Cancer Management. This schematic highlights key challenges and future research priorities in prostate cancer management, including overdiagnosis and overtreatment, emerging diagnostic tools, disparities in care, screening controversies, long-term treatment side effects, biomarker development, and the optimization of active surveillance protocols. Central themes such as risk stratification, targeted therapies, and optimized surveillance are emphasized as pivotal areas for innovation and improvement in patient outcomes (Created with BioRender.com. (RRID:SCR_018361), accessed 20 May 2025).

**Figure 2 ijms-26-05386-f002:**
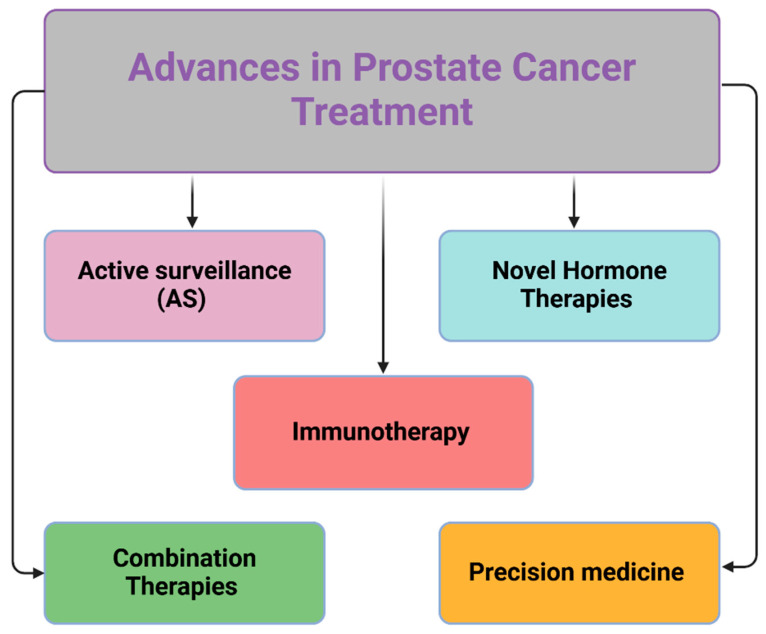
Advances in Prostate Cancer Treatment. This schematic illustrates five key contemporary approaches to prostate cancer management: Active Surveillance (AS) for low-risk disease, Novel Hormone Therapies targeting androgen signaling, Immunotherapy harnessing immune responses against cancer cells, Precision Medicine using targeted interventions based on molecular profiles, and Combination Therapies integrating multiple treatment modalities for enhanced efficacy (Created with BioRender.com. (RRID:SCR_018361), accessed 20 May 2025).

**Figure 3 ijms-26-05386-f003:**
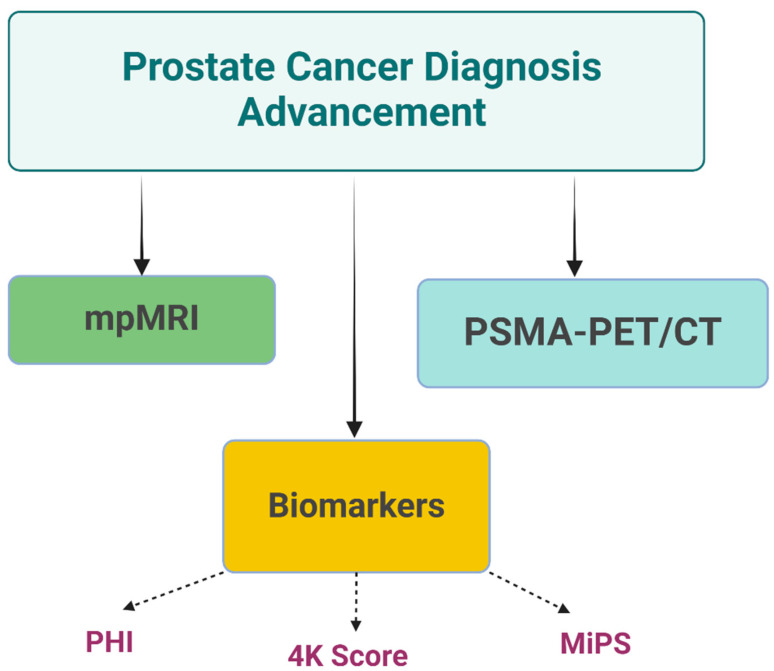
Schematic Overview of Advances in Prostate Cancer Diagnosis. This figure illustrates key components of modern prostate cancer diagnostics, highlighting the roles of multiparametric MRI (mpMRI), PSMA-PET/CT imaging, and biomarker assays (PHI, 4K Score, MiPS) in advancing detection and risk assessment (Created with BioRender.com. (RRID:SCR_018361), accessed 20 May 2025)).

**Figure 4 ijms-26-05386-f004:**
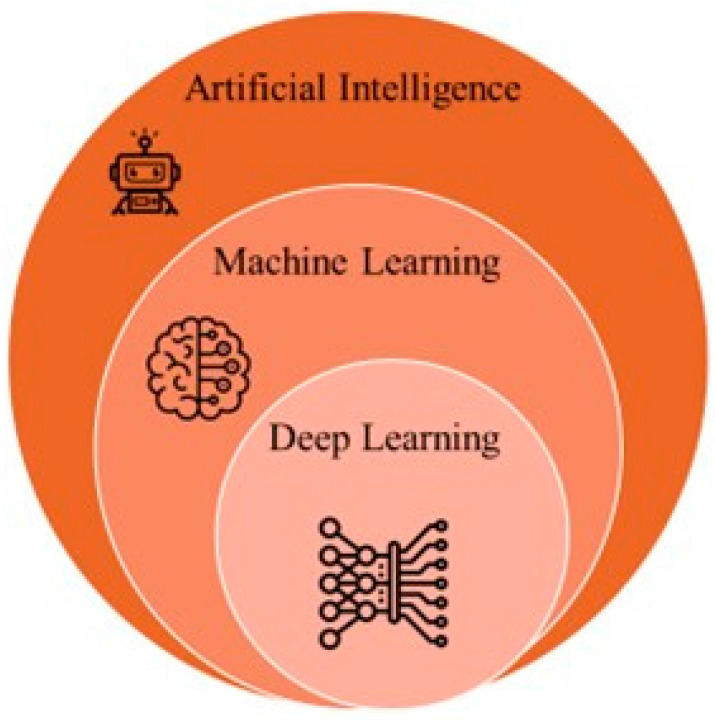
An Overview of AI, ML, and DL: Interconnections and Hierarchy (Created with BioRender.com. (RRID:SCR_018361), accessed 20 May 2025)).

**Figure 5 ijms-26-05386-f005:**
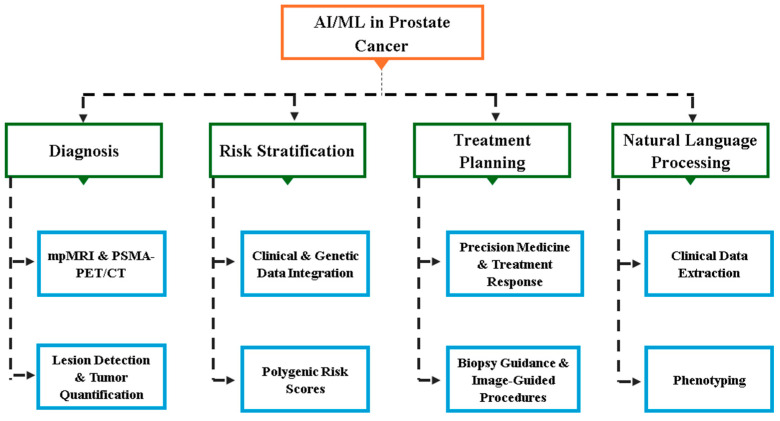
Hierarchical Overview of AI Applications in Prostate Cancer Management. This figure presents a hierarchical framework of artificial intelligence (AI) and machine learning (ML) applications across key domains of prostate cancer care, including diagnosis (mpMRI and PSMA-PET/CT for lesion detection and quantification), risk stratification (clinical/genetic data integration and polygenic risk scores), treatment planning (precision medicine, treatment response, and image-guided procedures), and natural language processing (clinical data extraction and phenotyping) (Created with BioRender.com. (RRID:SCR_018361), accessed 20 May 2025)).

**Figure 6 ijms-26-05386-f006:**
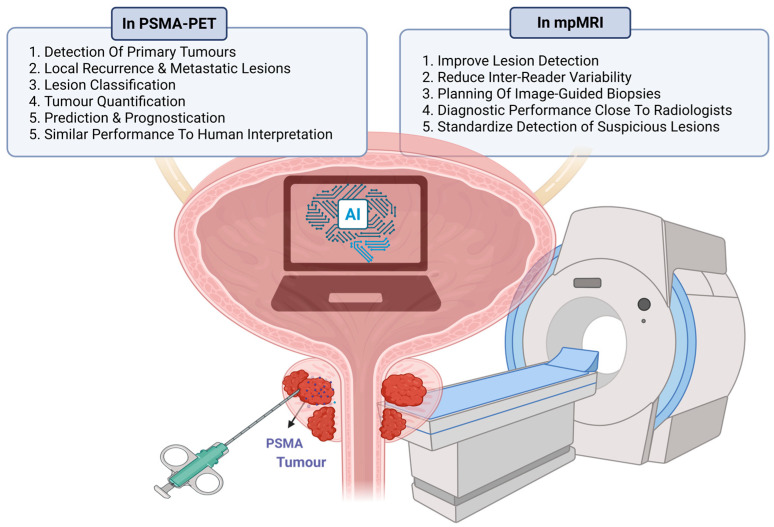
AI application in mpMRI and PSMA-PET. This figure illustrates the integration of artificial intelligence (AI) in multiparametric MRI (mpMRI) and PSMA-PET imaging for prostate cancer. In PSMA-PET, AI enhances primary tumor detection, lesion classification, quantification, prognostication, and provides performance comparable to human interpretation. In mpMRI, AI improves lesion detection, reduces inter-reader variability, aids in planning image-guided biopsies, offers diagnostic accuracy close to radiologists, and standardizes detection of suspicious lesions (Created with BioRender.com. (RRID:SCR_018361), accessed 20 May 2025)).

**Figure 7 ijms-26-05386-f007:**
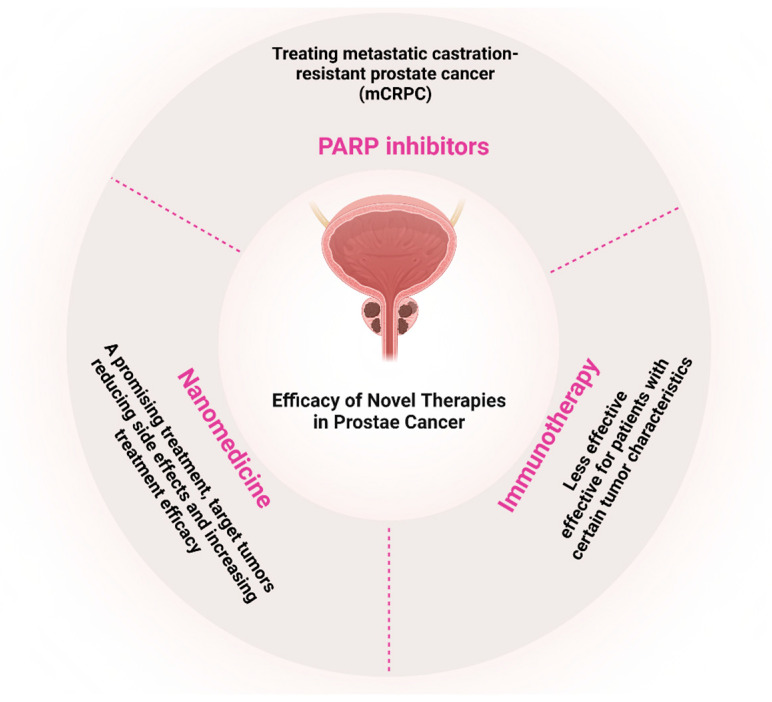
Efficacy and Key Features of Novel Therapies in PCa. This figure summarizes the roles of PARP inhibitors, immunotherapy, and nanomedicine in the treatment of metastatic castration-resistant PCa (mCRPC), highlighting their therapeutic potential, indications, and limitations (Created with BioRender.com. (RRID:SCR_018361), accessed 20 May 2025)).

**Figure 8 ijms-26-05386-f008:**
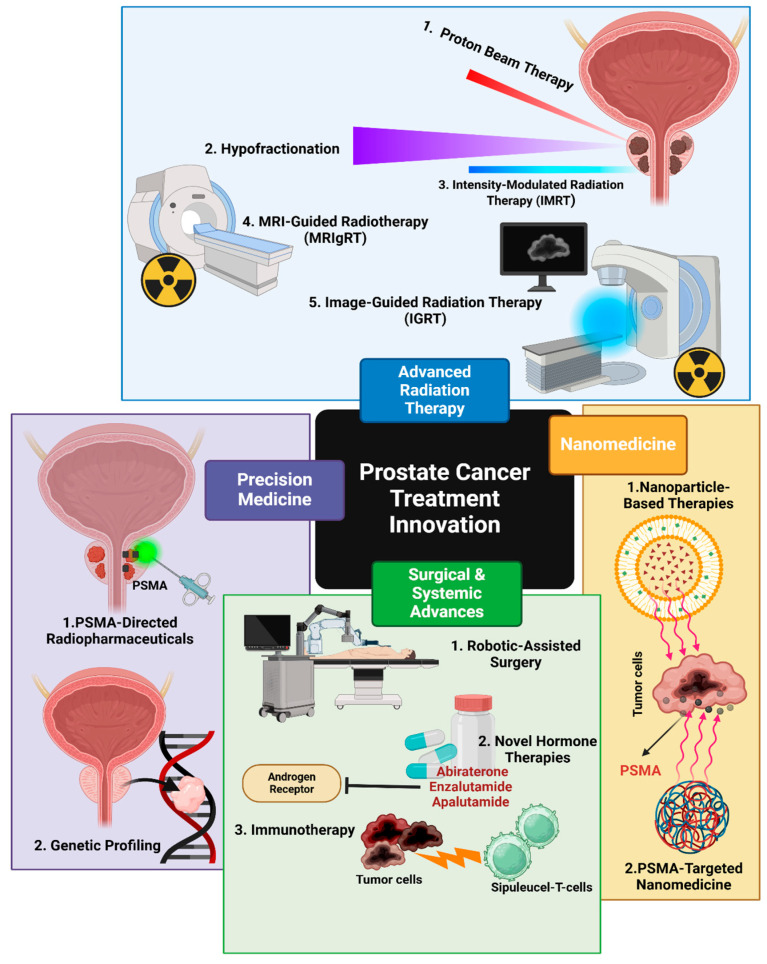
Innovations in PCa Treatment and Radiation Therapy. This figure highlights recent advancements in PCa management, emphasizing radiation therapy techniques. Four key innovation areas—Advanced Radiation Therapy, Surgical and Systemic Advances, Precision Medicine, and Nanomedicine—are illustrated in distinct colors for clarity (Created with BioRender.com. (RRID:SCR_018361), accessed 20 May 2025)).

**Figure 9 ijms-26-05386-f009:**
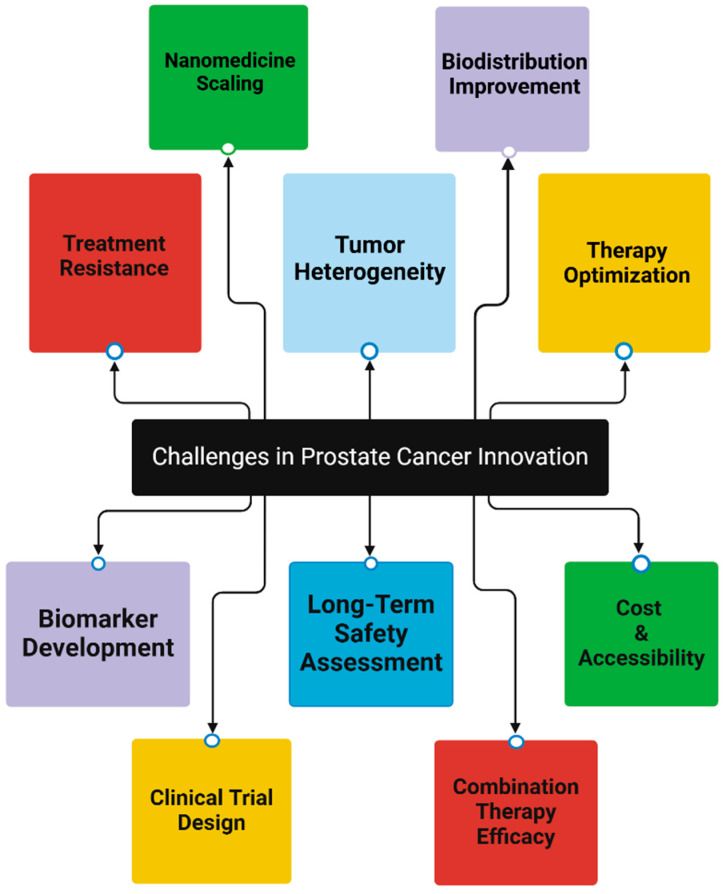
Flowchart Illustrating Key Challenges in PCa Treatment Innovation. The central node, “Challenges in PCa Innovation”, branches to 10 specific challenges: treatment resistance [124,125], tumor heterogeneity [126,127,128], Therapy optimization [125,129], Nanomedicine scaling [130,131], Biodistribution improvement [48,132], Biomarker development [133], Long-term safety assessment [134], Cost and accessibility [135], Clinical trial design [136], and Combination therapy efficacy [137] (Created with BioRender.com. (RRID:SCR_018361), accessed 20 May 2025)).

**Table 1 ijms-26-05386-t001:** AI and ML Applications in Prostate Cancer Management.

Application Area	Use Cases	Key Benefits	References
Imaging	-Lesion detection and classification on mpMR -Tumor segmentation and volume estimation -PSMA PET/CT image analysis -MRI–ultrasound fusion for targeted biopsies	-Improved accuracy and consistency in lesion detection -Reduced inter-reader variability -Automated quantification of tumor burden -Enhanced guidance for biopsy procedures	[79,80]
Predictive Modeling	-Risk stratification using clinical, genetic, and imaging data -Prediction of biochemical recurrence after treatment -Prediction of metastasis and treatment response	-More accurate prediction of clinically significant cancer -Personalized risk assessment -Improved patient counseling and treatment selection	[80,81,82]
Treatment Planning	-Automated organ-at-risk and target volume contouring -Dose optimization for radiation therapy -Prediction of optimal treatment modalities -Assessment of ADT benefits in combination with radiotherapy	-Increased efficiency in radiotherapy planning -More consistent treatment plans -Personalized treatment recommendations -Reduction of overtreatment	[79,83]

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
