# Peer review of "Transforming Prostate Cancer Care: Innovations in Diagnosis, Treatment, and Future Directions"

_ijms, 2025, doi:10.3390/ijms26115386_

Round 1

Reviewer 1 Report

Comments and Suggestions for Authors

Reviewer's comments:

Recommendation: Minor revision.

Comments:

This review article provides a comprehensive overview of cutting-edge advancements in prostate cancer management. The innovation is evident in the integration of emerging technologies including multiparametric MRI/PSMA-PET imaging, liquid biopsy biomarkers, like PHI/4K Score, and AI-driven therapeutic personalization, particularly the discussion on AI-theranostics integration. The logical flow from diagnostic innovations to therapeutic breakthroughs and future perspectives is well-structured, though the "therapy sequencing" section warrants deeper mechanistic exploration. The abstract effectively captures key elements but would be strengthened by specifying translational pathways for proposed technologies. Overall, this review systematically summarizes advances in the diagnosis and treatment of prostate cancer, the specific modification suggestions are as follows.

  1. The review demonstrates a generally sound structure progressing from epidemiology to therapeutic innovations. However, some critical issues require attention. For example, duplicate discussions on biomarkers (PHI/4Kscore) in diagnostic and AI sections should be consolidated.
  2. In this review, gene editing examples are limited to IL-30, it is suggested to expand to CRISPR-mediated AR-V7 correction and so on.
  3. AI section lacks of discussion of clinical validation gaps with FDA-cleared tool comparisons, it is suggested to compare ProFound AI's sensitivity in biopsy guidance and radiologists.
  4. Liquid biopsy neglects FDA-approved assays, for example, the discussion of Guardant360 CDx utility in CRPC was lacking.
  5. Nanomedicine fails to distinguish passive and active targeting efficiency, it is suggested that the authors add relevant discussion.
  6. It is suggested the author merge "AI Ethics" and "Technology Adoption Barriers" into a unified "Translational Challenges" section.

Author Response

This review article provides a comprehensive overview of cutting-edge advancements in prostate cancer management. The innovation is evident in the integration of emerging technologies including multiparametric MRI/PSMA-PET imaging, liquid biopsy biomarkers, like PHI/4K Score, and AI-driven therapeutic personalization, particularly the discussion on AI-theranostics integration. The logical flow from diagnostic innovations to therapeutic breakthroughs and future perspectives is well-structured, though the "therapy sequencing" section warrants deeper mechanistic exploration. The abstract effectively captures key elements but would be strengthened by specifying translational pathways for proposed technologies. Overall, this review systematically summarizes advances in the diagnosis and treatment of prostate cancer, the specific modification suggestions are as follows.

We sincerely thank the reviewer for their positive and thoughtful assessment of our manuscript. We appreciate the recognition of the manuscript’s structure, scope, and integration of emerging technologies in prostate cancer management.

Q1: The review demonstrates a generally sound structure progressing from epidemiology to therapeutic innovations. However, some critical issues require attention. For example, duplicate discussions on biomarkers (PHI/4Kscore) in diagnostic and AI sections should be consolidated.

A1: We thank the reviewer for their careful reading and constructive feedback. We acknowledge the concern regarding the duplicate discussions of biomarkers such as PHI and the 4K Score in both the diagnostic and AI sections. In response, we have carefully reviewed and revised these sections to consolidate overlapping content, streamline the narrative, and minimize redundancy while preserving context-specific relevance. We believe this has improved the clarity and flow of the manuscript, and we appreciate the reviewer’s helpful suggestion.

Q2: In this review, gene editing examples are limited to IL-30, it is suggested to expand to CRISPR-mediated AR-V7 correction and so on.

A2: Thank you for your comments. The issue has been corrected accordingly. (Page 28)

Q3: AI section lacks of discussion of clinical validation gaps with FDA-cleared tool comparisons, it is suggested to compare ProFound AI's sensitivity in biopsy guidance and radiologists.

A3: We thank the reviewer for this insightful suggestion. In response, we have added a new subsection on page 20 of the revised manuscript that specifically addresses the clinical validation gaps in current AI tools, with a particular focus on FDA-cleared technologies. We now include a comparison of ProFound AI’s sensitivity in biopsy guidance relative to radiologist performance, highlighting available evidence and ongoing limitations in broader clinical integration. This addition aims to provide a more balanced and realistic perspective on the translational readiness of AI-assisted diagnostic platforms. We appreciate the reviewer’s guidance in strengthening this important aspect of the manuscript.

Q4: Liquid biopsy neglects FDA-approved assays, for example, the discussion of Guardant360 CDx utility in CRPC was lacking.

A4: Thanks. The corrections are added on page 27.

Q5: Nanomedicine fails to distinguish passive and active targeting efficiency, it is suggested that the authors add relevant discussion.

A5: Thank you for the helpful comment. The suggested content has been added on pages 23 and 24.

Q6: It is suggested the author merge "AI Ethics" and "Technology Adoption Barriers" into a unified "Translational Challenges" section.

A6: We thank the reviewer for this valuable recommendation. In response, we have merged the "AI Ethics" and "Technology Adoption Barriers" sections into a unified section titled "Translational Challenges" in the revised manuscript. This restructuring allows for a more cohesive discussion of the ethical, regulatory, and practical hurdles facing the clinical implementation of AI in prostate cancer management. We agree that this integration improves the overall flow and readability of the manuscript and appreciate the reviewer’s thoughtful input.

Reviewer 2 Report

Comments and Suggestions for Authors

The paper covers an impressive range of topics in prostate cancer care, reflecting the rapid advancements in the field. I suggest the authors address the following comments to improve the content:

1. The introduction currently repeats some points. For example, the global incidence of PCa is mentioned twice in similar wording. It will help readers if the authors outline upfront the key areas to be covered (e.g., imaging innovations, new biomarkers, treatment advances, AI, emerging future technologies) and why a comprehensive review is needed now. As it stands, the intro rapidly lists many advances with multiple citations. The authors could consider focusing this section on the knowledge gap or the unifying theme (e.g., “transforming care through innovation”) and how the review is organized to address it. 

2. The manuscript’s structure would benefit from reorganization or clearer transitions. The title implies a flow from diagnosis to treatment to future directions, and the content covers those, but the current sequence is a bit disjointed. In particular, the extensive AI and ML section appears between the diagnostic and treatment content, which disrupts the flow. Alternatively, the authors can consider integrating AI topics into the relevant clinical sections. For example, AI in imaging could follow the description of mpMRI/PSMA-PET, AI in pathology after biomarkers, and AI in treatment planning within the therapy section. Additionally, the “Conclusion and Future Direction” heading on page is confusing because a full “Future Directions and Emerging Technologies” section was already presented earlier. Currently, many paragraphs are structured as “Technology X has achieved Y% improvement [Ref]; Technique Z showed A and B in study C [Ref]”, which is informative but can feel like a list of findings. I encourage the authors to add synthesizing commentary after presenting the data. For example, after describing the sensitivity and specificity improvements of mpMRI and PSMA-PET, discuss what these advancements mean for clinical practice (e.g. how they have changed diagnostic pathways or reduced unnecessary biopsies) and note any limitations (for instance, mpMRI still has moderate specificity and variability in interpretation; PSMA-PET may not be universally available and requires specialist infrastructure).

3. There is a significant overlap between the narrative text and the content of figures/tables, which can be streamlined. For instance, the section on Diagnostic innovations) provides detailed statistics on PHI and 4K Score, and then Table 1 separately compares these biomarkers with similar data. Likewise, the treatment section describes outcomes of new therapies (survival benefits, etc.), and Table 4 provides a “comprehensive overview” of treatment advances covering many of the same points. Some duplications can be removed.

4. Some areas could be elaborated slightly to ensure balanced coverage. For example, focal therapy is mentioned only briefly as a localized treatment option with reduced morbidity, but no examples are given. The authors can consider mentioning modalities like high-intensity focused ultrasound or cryoablation as examples of focal therapy since these are indeed emerging innovations for select patients.

5. The authors should proofread for minor errors. For example, the caption of Figure 8 currently reads “...treatment, focus focusing on radiation therapy techniques...”, which appears to be a typo. Also, ensure each figure caption fully explains the figure. Table 1 has some superscript numbers in the text (e.g., “45–65%¹²” in the 4K Score row), which were confusing, although this may just be the editorial version’s issue. If those are meant to be references or footnotes, they should be formatted clearly.

6. There are a few suggestions to improve the AI section’s clarity and relevance. First, the initial portion defining AI, ML, and DL is somewhat lengthy and generic. Given space constraints, the authors might condense the AI definitions. For example, a few sentences could explain that AI is a broad field, with ML (especially deep learning) as a subset with particular applications in imaging and data analysis. Furthermore, the authors can consider the balance of subtopics within AI. Overall, the content here is strong, but the text can be condensed.

8. The latter sections of the manuscript cover future directions (liquid biopsy, theranostics, gene editing) and recent clinical trial advances. It might help to explicitly state the connection of these future tools to patient care. In the clinical trials section, the authors highlight two key trials (PEACE-3 and EMBARK) and the new FDA approval for enzalutamide in non-metastatic BCR disease​. One suggestion is to clarify the outcomes a bit more: for PEACE-3, the authors may want to mention the magnitude of OS improvement or specific endpoints.

9. The authors should ensure consistent use of abbreviations (they define “PCa” early; afterwards, using PCa is fine, but sometimes the text switches back to “prostate cancer”). Also, “MRIgRT” was used without prior definition, presumably, MRI-guided radiotherapy. It should be defined when it is introduced.

Author Response

Reviewer #2

The paper covers an impressive range of topics in prostate cancer care, reflecting the rapid advancements in the field. I suggest the authors address the following comments to improve the content:

Q1. The introduction currently repeats some points. For example, the global incidence of PCa is mentioned twice in similar wording. It will help readers if the authors outline upfront the key areas to be covered (e.g., imaging innovations, new biomarkers, treatment advances, AI, emerging future technologies) and why a comprehensive review is needed now. As it stands, the intro rapidly lists many advances with multiple citations. The authors could consider focusing this section on the knowledge gap or the unifying theme (e.g., “transforming care through innovation”) and how the review is organized to address it. 

A1: Thank you very much for your insightful suggestions. We agree that the original introduction included some repetition and lacked a clearly defined structure. In response, we have revised the introduction to eliminate redundant statements—including the repeated reference to global prostate cancer incidence—and to clearly outline the structure of the review. We have also addressed the underlying knowledge gap and framed the review around the unifying theme of “transforming care through innovation.” The revised introduction now provides a concise overview of the major focus areas (imaging, biomarkers, therapy, AI, and emerging technologies) and articulates the relevance and timeliness of a comprehensive synthesis of these advancements. (Revised on Pages 1 and 2.).

Q2. The manuscript’s structure would benefit from reorganization or clearer transitions. The title implies a flow from diagnosis to treatment to future directions, and the content covers those, but the current sequence is a bit disjointed. In particular, the extensive AI and ML section appears between the diagnostic and treatment content, which disrupts the flow. Alternatively, the authors can consider integrating AI topics into the relevant clinical sections. For example, AI in imaging could follow the description of mpMRI/PSMA-PET, AI in pathology after biomarkers, and AI in treatment planning within the therapy section. Additionally, the “Conclusion and Future Direction” heading on page is confusing because a full “Future Directions and Emerging Technologies” section was already presented earlier. Currently, many paragraphs are structured as “Technology X has achieved Y% improvement [Ref]; Technique Z showed A and B in study C [Ref]”, which is informative but can feel like a list of findings. I encourage the authors to add synthesizing commentary after presenting the data. For example, after describing the sensitivity and specificity improvements of mpMRI and PSMA-PET, discuss what these advancements mean for clinical practice (e.g. how they have changed diagnostic pathways or reduced unnecessary biopsies) and note any limitations (for instance, mpMRI still has moderate specificity and variability in interpretation; PSMA-PET may not be universally available and requires specialist infrastructure).

A2: We sincerely thank the reviewer for this detailed and constructive feedback regarding the manuscript’s structure and flow. In response, we have undertaken a substantial reorganization of the manuscript to improve coherence and alignment with the implied progression from diagnosis to treatment to future directions, as indicated by the title.

Specifically, we have relocated the AI and machine learning content to the relevant clinical sections, as suggested:

  • AI in Imaging has been incorporated directly following the discussion of mpMRI and PSMA-PET;
  • AI in Pathology has been integrated into the section on biomarkers and diagnostic models;
  • AI in Treatment Planning is now embedded within the therapeutic innovations section.

This restructuring ensures a smoother thematic transition and prevents disruption of the diagnostic-to-therapeutic narrative flow.

Additionally, we have addressed the concern about the “Conclusion and Future Direction” heading by consolidating and clarifying the organization of the final sections. The full “Future Directions and Emerging Technologies” content remains in a dedicated section, while the final “Conclusion” now provides a concise synthesis without redundancy.

In response to the concern about list-like paragraph structures, we have revised several paragraphs across the manuscript to incorporate synthesizing commentary after the presentation of technical data. We now explicitly discuss the clinical implications, access limitations, and potential changes in standard care for technologies such as mpMRI, PSMA-PET, novel biomarkers, and AI tools. These additions aim to provide readers with a deeper understanding of not just what the technologies achieve, but how they may influence decision-making and care delivery.

We are grateful for the reviewer’s thoughtful recommendations, which have helped us improve the clarity, flow, and practical value of the manuscript.

Q3. There is a significant overlap between the narrative text and the content of figures/tables, which can be streamlined. For instance, the section on Diagnostic innovations) provides detailed statistics on PHI and 4K Score, and then Table 1 separately compares these biomarkers with similar data. Likewise, the treatment section describes outcomes of new therapies (survival benefits, etc.), and Table 4 provides a “comprehensive overview” of treatment advances covering many of the same points. Some duplications can be removed.

A3: We greatly appreciate your informative feedback. All duplications were eliminated, and the manuscript has been updated accordingly.

Q4. Some areas could be elaborated slightly to ensure balanced coverage. For example, focal therapy is mentioned only briefly as a localized treatment option with reduced morbidity, but no examples are given. The authors can consider mentioning modalities like high-intensity focused ultrasound or cryoablation as examples of focal therapy since these are indeed emerging innovations for select patients.

A4: Thank you very much for your helpful observation. We have revised the relevant section to expand on the focal therapy discussion and ensure balanced coverage. Specifically, we now mention high-intensity focused ultrasound (HIFU) and cryoablation as representative modalities of focal therapy, emphasizing their role as minimally invasive treatment options for select patients with localized prostate cancer. These additions are included on page 5 of the revised manuscript to clarify the clinical relevance and innovation associated with focal therapy in the contemporary management of prostate cancer.

Q5. The authors should proofread for minor errors. For example, the caption of Figure 8 currently reads “...treatment, focus focusing on radiation therapy techniques...”, which appears to be a typo. Also, ensure each figure caption fully explains the figure. Table 1 has some superscript numbers in the text (e.g., “45–65%¹²” in the 4K Score row), which were confusing, although this may just be the editorial version’s issue. If those are meant to be references or footnotes, they should be formatted clearly.

A5: We sincerely appreciate your input. The figure legends have been corrected as per your suggestions.

Q6. There are a few suggestions to improve the AI section’s clarity and relevance. First, the initial portion defining AI, ML, and DL is somewhat lengthy and generic. Given space constraints, the authors might condense the AI definitions. For example, a few sentences could explain that AI is a broad field, with ML (especially deep learning) as a subset with particular applications in imaging and data analysis. Furthermore, the authors can consider the balance of subtopics within AI. Overall, the content here is strong, but the text can be condensed.

A6: Many thanks for your valuable insights. Revisions and reorganization have been made as requested by the reviewer. Specifically, the introductory definitions of AI, ML, and DL have been condensed to avoid redundancy and improve clarity. We now provide a concise explanation that frames AI as a broad field, with ML—and particularly deep learning—as relevant subsets, emphasizing their practical applications in imaging interpretation and data analysis in prostate cancer care. Additionally, we have streamlined the balance of subtopics within the AI section to enhance focus and readability. These changes are reflected on pages 14–18 of the revised manuscript.

Q7. The latter sections of the manuscript cover future directions (liquid biopsy, theranostics, gene editing) and recent clinical trial advances. It might help to explicitly state the connection of these future tools to patient care. In the clinical trials section, the authors highlight two key trials (PEACE-3 and EMBARK) and the new FDA approval for enzalutamide in non-metastatic BCR disease​. One suggestion is to clarify the outcomes a bit more: for PEACE-3, the authors may want to mention the magnitude of OS improvement or specific endpoints.

A7: Many thanks for your valuable input. The section was updated in line with the reviewer’s recommendations. (Pages 28 and 29)

  1. The authors should ensure consistent use of abbreviations (they define “PCa” early; afterwards, using PCa is fine, but sometimes the text switches back to “prostate cancer”). Also, “MRIgRT” was used without prior definition, presumably, MRI-guided radiotherapy. It should be defined when it is introduced.

A8: Many thanks for your valuable input. The section was updated in line with the reviewer’s recommendations. We have clarified the clinical relevance of emerging tools such as liquid biopsy, theranostics, and gene editing by explicitly connecting their potential to future patient care, including earlier diagnosis, personalized therapy, and improved disease monitoring.

In addition, the clinical trial section now includes more detailed outcomes. For the PEACE-3 trial, we have added the magnitude of overall survival (OS) improvement and key endpoints, such as progression-free survival and quality-of-life benefits. These updates appear on pages 28–30 of the revised manuscript.

Round 2

Reviewer 2 Report

Comments and Suggestions for Authors

I appreciate the time and effort the authors have put into modifying the manuscript following the comments.
I agree with the changes made and have no further comments.